

# Do the effects of crops on skylark (*Alauda arvensis*) differ between the field and landscape scales?

Christophe Sausse[1,2,3], Aude Barbottin[2,3], Frédéric Jiguet[4] and Philippe Martin[2,3]

[1] Terres Inovia, Thiverval-Grignon, France
[2] AgroParisTech, UMR 1048 SAD-APT, Thiverval-Grignon, France
[3] INRA, UMR 1048 SAD-APT, Thiverval-Grignon, France
[4] Centre d'Ecologie et des Sciences de la Conservation UMR7204 CNRS-MNHN-UPMC-Sorbonne Universités, Paris, France

## ABSTRACT

The promotion of biodiversity in agricultural areas involves actions at the landscape scale, and the management of cropping patterns is considered an important means of achieving this goal. However, most of the available knowledge about the impact of crops on biodiversity has been obtained at the field scale, and is generally grouped together under the umbrella term "crop suitability." Can field-scale knowledge be used to predict the impact on populations across landscapes? We studied the impact of maize and rapeseed on the abundance of skylark (*Alauda arvensis*). Field-scale studies in Western Europe have reported diverse impacts on habitat selection and demography. We assessed the consistency between field-scale knowledge and landscape-scale observations, using high-resolution databases describing crops and other habitats for the 4 km$^2$ grid scales analyzed in the French Breeding Bird Survey. We used generalized linear models to estimate the impact of each studied crop at the landscape scale. We stratified the squares according to the local and geographical contexts, to ensure that the conclusions drawn were valid in a wide range of contexts. Our results were not consistent with field knowledge for rapeseed, and were consistent for maize only in grassland contexts. However, the effect sizes were much smaller than those of structural landscape features. These results suggest that upscaling from the field scale to the landscape scale leads to an integration of new agronomic and ecological processes, making the objects studied more complex than simple "crop ∗ species" pairs. We conclude that the carrying capacity of agricultural landscapes cannot be deduced from the suitability of their components.

# INTRODUCTION

Actions favoring biodiversity in agricultural areas in Europe have been inspired by the principle of "wildlife-friendly farming," also known as "land-sharing" between farmers and heritage and common species (*Green et al., 2005*). These actions constitute a win–win strategy, in which conservation goals are met and economic profit is achieved, through

Corresponding author
Christophe Sausse,
c.sausse@terresinovia.fr

ecosystem services (*Tscharntke et al., 2012*). Within such strategies, cropping patterns are an important means of improving landscape quality (*Benton, Vickery & Wilson, 2003*; *Tscharntke et al., 2005*). Indeed, crops serve as a habitat for a number of species and the loss of diversity resulting from agricultural intensification is considered to have been a major component of biodiversity loss in Europe (*Donald, Green & Heath, 2001a*). Moreover, cropping patterns are designed at the landscape scale, which is more appropriate than the field scale for the assessment and preservation of biodiversity (*Burel et al., 1998*; *Tscharntke et al., 2005*). They are more labile than fixed landscape elements, as they change every year due to crop rotation and over periods of several years under the influence of market forces and public policies.

Farmland birds may be considered a good surrogate for agricultural landscape quality (*Gregory et al., 2005*). Knowledge about the impact of crops on these birds would therefore facilitate more effective action to preserve biodiversity at the farm scale and beyond. However, most of the available knowledge relates to individuals in their immediate environment: a cultivated field or a spot corresponding to a detection area (about 100 m around the observer). Crops provide various trophic resources for birds (*Holland et al., 2012*). Moreover, their structure can affect nesting and protection against predators (*Wilson, Whittingham & Bradbury, 2005*). These effects are subtle and may vary across seasons. For example, skylarks (*Alauda arvensis*) prefer to forage in winter in high rather than low cereal stubbles, which indicates a cryptic strategy against predators (*Butler, Bradbury & Whittingham, 2005*). But *Powolny et al. (2014)* showed that this behavior was more common among females than males, which preferred flight. In contrast, high and dense crops, like winter cereals, are poorly selected during the breeding season (*Donald et al., 2001b*). Their rapid growth limits the number of nesting attempts, although it can mitigate the impact of predation, with a global negative impact on productivity (*Donald et al., 2002*). This example shows two ecological processes at work in crops: habitat selection, which is a behavioral process, and population increase rate thanks to resources, food and protection provided by the crop. These ecological processes are often translated into terms of crop suitability for nesting and foraging. This concept could be used directly to explain the overall decline of farmland bird populations, as a result of farming management regimes (e.g., switch from spring- to winter-sown crops (*Chamberlain, Vickery & Gough, 2000*)), or indirectly as model parameters for the assessment of a land-use scenario (*Boatman et al., 2010*; *Topping, Odderskær & Kahlert, 2013*; *Everaars, Frank & Huth, 2014*; *Brandt & Glemnitz, 2014*). The rationale underlying these approaches is that the carrying capacity of the agricultural landscape, considered in a general sense as the density that can be sustained for a long period of time (*Dhondt, 1988*), is the addition of the carrying capacities of its components: the crops considered as habitats. We aimed to test this hypothesis. Can field-scale knowledge about crop suitability be used to predict the impact on populations of farmland birds across landscapes?

We chose skylark as the model species for this study because considerable amounts of information about crop suitability in Western Europe are available for this species (e.g., *Wilson et al., 1997*; *Donald, Green & Heath, 2001a*; *Donald et al., 2001b*; *Eraud &*

*Boutin, 2002*, for the breeding period). This species remains very common, but its numbers have recently declined and its characteristics as an open countryside specialist make it an interesting model for studies of the impact of agriculture management on biodiversity. We focused on the breeding period, when skylark shows a territorial behavior and may nest in crops. Previous studies have reported a general positive association between some crops and skylarks during this period. For example, *Eraud & Boutin (2002)* showed that skylark nest density was highest in alfalfa and set-aside in South-West France, *Chamberlain et al. (1999)* observed a similar trend for set-aside in England, over 1 km$^2$ landscapes. *Wilcox et al. (2014)* showed that more skylark territories could be found in set-aside or in legumes (including bean, pea and alfalfa crops) than in other crops. By contrast, other crops, such as rapeseed and maize in particular, seem to have a negative impact on skylark densities. These two widespread crops have contrasting cropping cycles: August to July for rapeseed, and April to October for maize, in most French contexts. The skylark nests on the ground and is most comfortable when the vegetation is short. This species would therefore be expected to be disadvantaged by rapeseed and maize, which are among the tallest annual crops.

Field-scale studies in western France showed that skylark selected rapeseed less frequently for nesting than other crops (*Eraud & Boutin, 2002*; *Miguet, Gaucherel & Bretagnolle, 2013*). *Whittingham, Wilson & Donald (2003)* drew the same conclusion for two of three regions of the UK studied, accounting for the positive effect of rapeseed in the remaining region by late crop establishment in the fields sampled. *Chamberlain et al. (1999)* showed that the probability of skylark occupancy was lower for rapeseed than for winter cereals. *Eraud & Boutin (2002)* found that rapeseed decreased the breeding success of skylarks. *Wilson et al. (1997)* noted that skylarks could establish territories within rapeseed crops, but without nesting, which was hampered by the rapid development of this crop and accordingly an unsuitable vegetation structure. However, *Siriwardena, Cooke & Sutherland (2012)* showed, for 1 km$^2$ landscapes, that there was a positive or neutral association (depending on the control variables) between skylark density and rapeseed in the lowland context, confirming the positive association found by *Chamberlain & Gregory (1999)* for the early breeding season only. The impact of maize has been less thoroughly studied, as this crop is relatively rare in the UK, where many of the studies on farmland birds were carried out. *Eraud & Boutin (2002)* showed that maize had a negative effect on the density of skylark territories. *Dziewiaty & Bernardy (2007)* drew the same conclusion in Germany, and they considered maize to be an ecological trap whose rapid growth hampered the detection of predators. Recent studies on the impact of bioenergy crops in Germany used scores of crop suitability for nesting and feeding, obtained from previous studies, as model parameters (*Everaars, Frank & Huth, 2014*; *Brandt & Glemnitz, 2014*). Both these studies considered rapeseed crops to be unsuitable for both the nesting and feeding of skylarks. Maize was considered unsuitable for nesting in both studies, but one of these studies (*Brandt & Glemnitz, 2014*) considered it to be suitable for feeding, whereas the other (*Everaars, Frank & Huth, 2014*) did not. All these references concern various farming contexts in the UK, Germany and France, which are largely comparable. Rapeseed

is a component of crop rotations dominated by cereals giving rise to stubble, and maize can be cultivated in monoculture. However, the crop cycle and subsequent management of the intercropping period may differ slightly between latitudes. With few exceptions, the studies carried out did not mention the agricultural practices or conditions likely to generate subtle differences in crop structure or food resources (e.g., fertilization, soil tillage).

In summary, most field-scale studies have suggested that the overall suitability of rapeseed and maize for skylark is low. A constant effect of these crops at the field and landscape scales would therefore imply that the carrying capacity of the landscape would be decreased by the presence of large areas under these crops. However, landscape-scale studies in the UK have cast doubts on this hypothesis in the case of rapeseed.

We tested the hypothesis of invariant effects in the French context, on larger landscapes of 4 km$^2$, making use of the variation of crop composition between the grid squares of the French Breeding Bird Survey (FBBS). This 4 km$^2$ scale is much larger than skylarks' territories and may potentially accommodate several dozen couples, according to a maximum of 3.3–3.7 territories by 10 ha found by *Eraud & Boutin (2002)*. It is a manageable landscape mosaic from the farmer's point of view. We used nested models to estimate the response of skylark abundance to variations of rapeseed and maize areas between squares and to assess the consistency of effects between the field and landscape scales, by checking the signs of correlation coefficients. According to our hypotheses, we expected lower densities of skylarks in landscapes where maize or rapeseed areas were high. *Whittingham et al. (2007)* and *Schaub et al. (2011)* showed that the habitat-density associations identified for farmland birds in one region did not necessarily applied to other regions, in the UK and Switzerland, respectively. We studied the effects of rapeseed and maize on skylark densities throughout France, stratifying landscapes according to local and geographic contexts, to ensure that our conclusions were valid for a large range of contexts.

# MATERIALS AND METHODS

## Bird data

We used data from the French Breeding Bird Survey (FBBS), a monitoring program in which volunteer skilled ornithologists count birds following a standardized protocol at the same plot, each year since 2001 (*Jiguet et al., 2012*). Each year, species abundances were recorded in each 2 km × 2 km squares whose centroids were located within a 10 km radius around a locality specified by the volunteer. On each plot, volunteers carried out ten point counts (5 min each, separated by at least 300 m) twice per spring within three weeks around the pivotal date of May 8th to ensure the detection of both early and late breeders. To be validated, counts must be repeated at approximately the same date between years (±7 days) and at dawn (within 1–4 h after sunrise) by a unique observer in the same order. The maximum count per point for the two spring sessions was retained as an indication of point-level species abundance. The counts obtained at the 10 points were summed to give the abundance for the entire square. The FBBS focuses on common birds that regularly breed in France, hence monitors the breeding skylark across the country.

**Table 1 Landscape descriptors.**

| Variable | Source |
| --- | --- |
| **Fixed elements** | |
| In agricultural areas | |
| Annual crop area | LPIS |
| "Grass" area, i.e., permanent crops, mostly grass and alfalfa | LPIS |
| Arboriculture and vineyard area | LPIS |
| Tree area (hedgerows, groves) | BD Topo® vegetation layer |
| Agricultural areas not belonging to any of the above classes (corresponding to interstitial areas, such as field margins, pathways, small buildings, etc.) | All Corine Land Cover classes "Agriculture" not belonging to the LPIS and BD Topo® vegetation layer |
| Number of cropping blocks | LPIS |
| Number of distinct tree patches | BD Topo® vegetation layer |
| In non-agricultural areas | |
| Artificialized area | Corine Land Cover |
| Wetland area | Corine Land Cover |
| Free water area | Corine Land Cover |
| Herbaceous and shrubby areas | Corine Land Cover |
| Forest area | Corine Land Cover |
| Road length | |
| Length of non-asphalted road | BD Topo® road layer |
| Length of road with low traffic levels | BD Topo® road layer |
| Length of road with high traffic levels | BD Topo® road layer |
| **Annual crops (nested in annual crop area)** | |
| Maize area | LPIS |
| Rapeseed area | LPIS |
| Cereal area (wheat, barley, other stubble cereals, both winter and spring types) | LPIS |

**Notes.**

See the glossary for definitions.

LPIS, Land Parcel Identification System; CAP, Common Agricultural Policy.

## Landscape data

For the identification of landscape factors affecting farmland birds, we carried out a literature review based on studies using data from French and UK breeding bird surveys (*Chamberlain & Gregory, 1999*; *Devictor & Jiguet, 2007*; *Siriwardena, Cooke & Sutherland, 2012*) or studies focusing on single factors, such as roads (*Reijnen, Foppen & Meeuwsen, 1996*). The variables used in this study are shown in Table 1. These variables were obtained from three national databases: the Land Parcel Identification System (LPIS) 2007–2010, used for the administration of the Common Agricultural Policy (CAP), the BD topo® from *Institut Géographique National*, and Corine Land Cover 2006 (Table 2).

These geographic data were integrated into a single database, with priority given to the data with the best spatial resolution: the BD topo®, followed by the LPIS and finally Corine Land Cover, mostly to cover the gaps in non-agricultural areas.

The French LPIS is not spatially explicit at crop level. It focuses on cropping blocks composed of one or several fields. Each block is a polygon, the attributes of which are the areas covered by the crops within it, with no specific information provided about

Sausse et al. (2015), *PeerJ*, DOI 10.7717/peerj.1097

**Table 2  National databases used to describe the landscape covering the FBBS squares.**

| Database | Spatial objects | Attributes | Time interval | Planimetric accuracy | Source | Provider |
|---|---|---|---|---|---|---|
| Land Parcel Identification System 2007–2010 | Polygons corresponding to at least one field with annual or permanent or ligneous crops | Crops (28 classes) and their area in each polygon | Each year | A few meters | Declaration by farmers | *Agence de Services et de Paiements*<br><br>http://www.asp-public.fr |
| CORINE Land Cover 2006 | Polygons | 44 land cover classes | 2006 ± 1 year | Less than 100 m | Satellite | European Environment Agency<br>http://www.eea.europa.eu |
| BD Topo®, vegetation and road layers | Polygons (vegetation) and polylines (roads) | 1 class for trees<br><br>5 classes for roads | Between 1999 and 2007 | 5 m | Orthophotography | *Institut Géographique National*<br><br>http://www.ign.fr |

the location of each crop within the block. It was not, therefore, possible to calculate indicators of crop configuration, and estimates of crop area were imprecise when the blocks intersected with FBBS squares. We resolved this problem by considering the area under a crop within such blocks to be equal to the area of the block within the square multiplied by the proportion of the crop in the block. The LPIS did not distinguish between spring and winter crops. Both winter and spring rapeseed crops were present, but this was of very little consequence because the spring type was largely underrepresented (0.2% of the area under rapeseed in France for 2007–2010, *French Ministry of Agriculture, 2009*). The "industrial set-aside" category of the LPIS was considered to correspond to rapeseed, based on cross-checking with data for the administrative area (*French Ministry of Agriculture, 2009*).

The relationships between birds and crops studied here may involve multiple ecological processes: the selection of the squares by skylarks in the year of observation, but also the demographic advantage or disadvantage conferred by the quality of the habitats within these squares. We tried to isolate this last term, to identify long-term effects on the carrying capacity of the landscape regardless of inter-annual crop variations. The four-year study period was too short to take large changes in cropping systems into account. We therefore pooled the data and used average values for both abundance and crop composition, for single squares followed for more than one year between 2007 and 2010.

## Sample selection and landscape stratification

We initially selected the FBBS squares for 2007–2010, as LPIS data were available for the corresponding period. We then restricted the study to agricultural contexts, by selecting squares with more than 50% of their area under agriculture according to the LPIS.

According to *Whittingham et al. (2007)*, habitat-density associations may be dependent on landscape type (e.g., openfield vs. grassland), bird density, and geographic context, with this last factor being the most important. We therefore stratified the FBBS squares as a function of landscape type and ecological region, as given by the digital map of European ecological regions (DMEER version 2003) from the European Environment Agency.

Arable crops, grass and trees in agricultural areas strongly influenced the abundance of skylarks (*Chamberlain & Gregory, 1999*; *Robinson, Wilson & Crick, 2001*). We therefore stratified the FBBS squares according to the grass and tree factors, with an indirect inclusion of arable crops, due to high correlation with grass ($-0.87$). FBBS squares were classified according to their position on either side of the curve defined by an equation, the parameters of which were selected so as to give equal weightings to both criteria according to their different ranges of variation, and to obtain two well-balanced groups:

$$\sqrt{(0.75 * \text{grass area})^2 + (2 * \text{tree area})^2} = 100. \tag{1}$$

The group below the curve was called "open-field," and the group above was referred to as "grassland" (Fig. 1).

The European ecological regions data incorporate information about climate, flora and topography. Some of these regions contain only marginal parts of France. We therefore

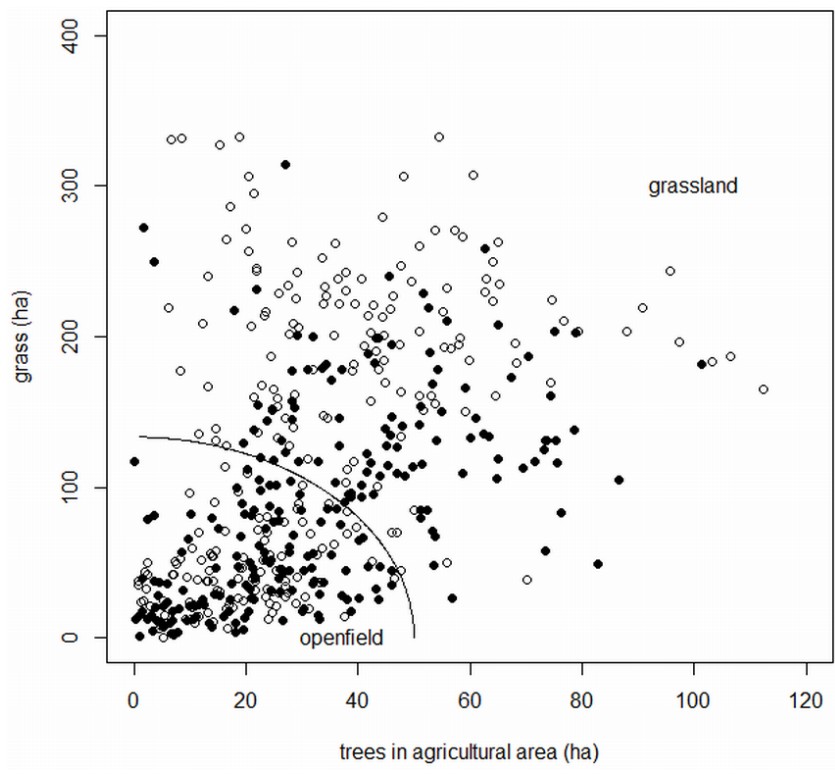

**Figure 1 Stratification of the squares.** "Openfield" and "grassland" on both sides of the curve defined by the Eq. (1) given in the text; closed circles: Southern temperate Atlantic ecoregion ("West"); open circles: Western European broadleaf forest ecoregion ("East").

retained only the "Southern temperate Atlantic" and "Western European broadleaf forest" regions, which together include 97% of the previously selected FBBS squares, and which split France into two roughly equal parts, corresponding to the West and the East (Fig. 2). The limit between ecological regions was approximated on the basis of administrative zones. Cross-referencing of the two stratifications yielded four groups: Openfield East; Openfield West; Grassland East; Grassland West.

Once the squares had been assigned to these four groups, we eliminated those considered potentially unsuitable for the crop of interest, by retaining the squares in which its area was non-zero. All squares were considered potentially suitable for skylark according to the large range of this species and the presence of favorable agricultural habitats. We eventually obtained eight samples, corresponding to four groups ∗ two factors of interest (the rapeseed and maize areas; Table 3).

## Statistical analysis

Crop compositions are constrained by agronomic rules. For example, rapeseed is systematically grown in rotations with cereals, to the benefit of both species, as this approach improves weed and pest management. Successive rapeseed crops are usually separated by at least three years in the rotation (e.g., rapeseed followed by wheat and barley before a return to rapeseed). Even in landscapes dominated by such a short rotation, the

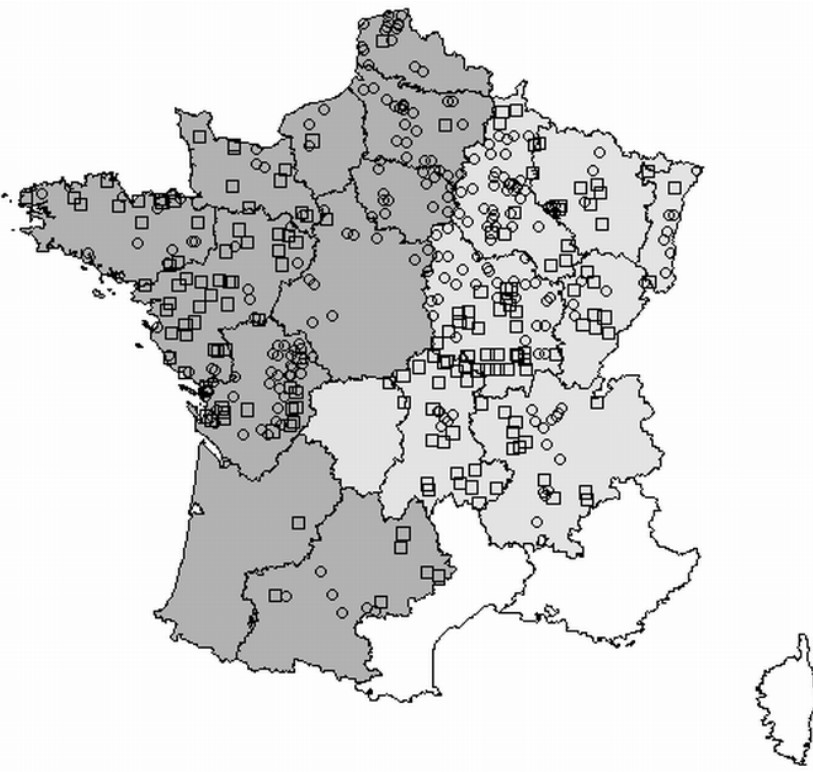

**Figure 2  Map of the survey squares.** Circles, open-field; squares, grassland; dark gray, Southern temperate Atlantic ecoregion ("West"); light gray, Western European broadleaf forest ecoregion ("East"); black lines, limits between administrative regions.

area under rapeseed therefore cannot exceed one third of the total area under annual crops. By contrast, maize can be cultivated either in rotations or as a monoculture; its area is therefore not limited. These structural relationships may make it difficult to establish isolated responses to individual crops. Confounding effects may occur between crops, or between crops and the total area under annual crops or grass. Before investigating responses, we checked the correlations between these variables for each square sample.

We estimated the relationships between skylark abundance and rapeseed of maize areas for the various squares according to an information theoretic approach. We first built three nested general linear models, where abundance depended on: (1) an autocovariate to minimize the effects of the spatial autocorrelation of abundances (*Augustin, Mugglestone & Buckland, 1996*), (2) the autocovariate, and the set of fixed elements listed in Table 1, but without the forest area (i.e., used of the whole set would generate collinearity due to the sum of areas being equal to 400 ha), (3) the autocovariate, the set of fixed elements, and the tested factor, i.e., rapeseed or maize area. We used negative binomial regressions due to overdispersion of the count data. Then we considered for each model and all the possible combinations of predictors. The resulting models were compared with Akaike information criterion (AICc with small sample size correction), and we used model-averaging to calculate parameter estimates and 95% confidence intervals for the top

Sausse et al. (2015), *PeerJ*, DOI 10.7717/peerj.1097

**Table 3 Description of the samples used to estimate the responses of skylarks to rapeseed and maize crop areas.**

| | Factor | Rapeseed area (ha) | | | | Maize area (ha) | | | |
|---|---|---|---|---|---|---|---|---|---|
| | Group | Openfield east | Openfield west | Grassland east | Grassland west | Openfield east | Openfield west | Grassland east | Grass-land west |
| **Sample** | Number of squares | 107 | 134 | 70 | 80 | 91 | 139 | 120 | 98 |
| | Variation of the factor | <1–103.7 | <1–82 | <1–67 | <1–45 | <1–315 | <1–173 | <1–118 | 1–112 |
| | Factor/annual crop area: maximum (%) | 39 | 33 | 34 | 26 | 84 | 71 | 100 | 85 |
| | Variation of annual crop area (ha) | 119–368 | 76–387 | 10–225 | 9–240 | 119–356 | 76–387 | <1–225 | 6–240 |
| | Skylark abundance (median–maximum) | 16–53 | 11–37 | 6–43 | 3–32 | 14–53 | 11–62 | 3–43 | 3–41 |
| **Correlation** | Annual crop area | 0.33 | 0.41 | 0.67 | 0.50 | 0.14 | −0.18 | 0.61 | 0.45 |
| | Grass area | −0.18 | −0.42 | −0.52 | −0.42 | −0.21 | 0.35 | −0.45 | −0.30 |
| | Rapeseed area | / | / | / | / | −0.63 | −0.35 | 0.09 | −0.26 |
| | Maize area | −0.62 | −0.37 | −0.03 | −0.31 | / | / | / | / |
| | Cereal area | 0.67 | 0.41 | 0.71 | 0.58 | −0.68 | −0.46 | 0.32 | −0.02 |

models ($\Delta$AICc < 2). The influence of sampling on the results was assessed by repeating the analysis 100 times on two third of the data.

### Implementation

Data were input and managed with the PostgreSQL 9.2.4 relational database server and its spatial extension PostGIS. The choice of this software was based on its ability to handle entire national databases. The statistical analyses were performed with R version 3.0.1, and the 'spdep' 'MASS' and 'MuMin' packages.

## RESULTS

Our samples cover wide ranges of variation representative of French agricultural contexts (Table 3). The open-field groups were, as expected, dominated by annual crops. The maximum crop proportions were consistent with the expert agronomic predictions. Maize covered the entire area under annual crops in some squares, whereas rapeseed area only exceeded one third of the total area under annual crops in one case, possibly due to a discrete field size effect. The correlations between crops were as expected. Indeed, rapeseed was associated with cereals and not with grass, and a spatial exclusion was observed in openfield contexts between maize on one hand and cereals and rapeseed on the other. However, the stringency of the correlations observed depended on the group to which the square concerned belonged.

We highlighted differences in the responses to the factors tested (Table 4) according to regional and local context. According to the parameter confidence intervals, the responses to maize were negative in both grassland contexts. The responses to rapeseed were null or, positive only in the grassland west context. However, the bootstrap procedure showed that the positive response to rapeseed in the grassland west context was less reliable than the negative responses to maize.

The correlations between crops provided information about possible confusion due to the coherence of the cropping systems (Table 3). The weak positive rapeseed-cereals (0.41) and rapeseed-annual crop (0.41) correlations observed in the open-field west context indicated a low level of spatial association, consistent with a low likelihood of confounding effects. By contrast, these spatial associations were stronger in the grassland contexts (east: 0.71 and 0.67; west: 0.50 and 0.58), in which confounding effects were considered more plausible. We did not find a spatial exclusion between maize and cereals in grassland west (−0.02) or east (0.32) that could have explained the negative responses to maize in these contexts.

The regression coefficients (Table 4) indicated that the studied factors had low effect sizes, at most 0.03 birds more or less per ha of rapeseed or maize. A comparison of AICcs suggested that the factors studied had a weaker influence than landscape elements. For example in grassland west, the addition of fixed elements to the autocovariate decreased the AICc by 5% (rapeseed) or 3% (maize), whereas the addition of rapeseed or maize decreased the AICc by 1% in both cases.

**Table 4 Results of the analysis of the response of skylark abundance to rapeseed and maize areas.**

| Factor | Group | Abundance ~ autocovariate AICc | Abundance ~ fixed elements + autocovariate Top model AICc | Abundance ~ fixed elements + factor + autocovariate Top model AICc | Coefficient of the factor Lower confidence interval | Upper confidence interval | Sampling influence (100 random samples on the 2/3) % lower confidence intervals >0 | % upper confidence intervals >0 |
|---|---|---|---|---|---|---|---|---|
| Rapeseed area (ha) | Openfield east | 795.9 | 758.8 | 758.8 | −0.003 | 0.007 | 2 | 100 |
| | Openfield west | 906.3 | 867.7 | 867.7 | −0.001 | 0.009 | 27 | 100 |
| | Grassland east | 430.2 | 416.5 | 416.5 | / | / | / | / |
| | Grassland west | 449.9 | 428.9 | 425.0 | 0.007 | 0.049 | 67 | 100 |
| Maize area (ha) | Openfield east | 651.4 | 620.8 | 620.8 | / | / | / | / |
| | Openfield west | 959.6 | 912.2 | 912.2 | / | / | / | / |
| | Grassland east | 633.5 | 608.7 | 590.2 | −0.052 | −0.024 | 0 | 2 |
| | Grassland west | 535.5 | 520.8 | 515.3 | −0.021 | −0.005 | 0 | 17 |

**Notes.**

/, factor not retained in the top models.

# DISCUSSION

Our study highlighted the lack of consistency between the responses of skylark populations at the landscape and field scales. Rapeseed was considered to have a low suitability for skylarks, but our analyses revealed a positive response to this crop in one context. The responses to maize and were partially consistent with expectations based on field-scale data, with the expected negative effects occurring only in grassland contexts. However, our results must be considered in a cropping system perspective. The positive response to rapeseed in grassland west context could not completely be distinguished from that to cereals, due to correlation between these variables. These results were supported in part by the results previously obtained in UK lowland areas by *Siriwardena, Cooke & Sutherland (2012)*, showing a positive response of skylark abundance on 1 km$^2$ landscapes to rapeseed area, conditionally to landscape structure or (but not and) field boundaries. This study was however conducted on smaller landscapes on only one year.

The range of variation explored in this study was very large and close to that experienced in the field, due to the large number of squares considered. Are these conditions likely to change in the near future, with a potential impact on the phenomena studied? We consider this to be unlikely for crop rotations. Shortening the interval between successive rapeseed crops in the rotation, leading to an increase in the maximum area under this crop, is not currently on the agenda for agronomic reasons, as this would hinder weed management. However, some innovations could probably change the suitability of crops as habitat for birds. For example, the use of GM rapeseed varieties would change food resources according to the Farm Scale Evaluation study (*Squire et al., 2003*), and the use of associated cover crops, such as legumes, would change both crop structure and food

resources. However, we consider it unlikely that these innovations will be extended to cover large areas in France in the near future.

## Origins of the discrepancies between the field and landscape scales

Our results suggest that field-scale studies do not take agronomic and ecological mechanisms operating at larger spatial and temporal scales into account, which is consistent with ecological theory: upscaling involves moving to higher levels of biological organization and larger spatiotemporal extents. This increases complexity and tends to decrease the generality of ecological findings (*Lawton, 1999*). Diverse biotic and abiotic interactions within the landscape may exacerbate or mitigate impacts. For example, the benefit of organic farming is smaller at farm level than at field level according to *Bengtsson, Ahnström & Weibull (2005)*.

In our case, the discrepancies between field and landscape may be accounted for by mitigation due to the diluted impact of the crop in landscapes to which other habitats make a major contribution. For rapeseed, the constraints on crop rotation have a strong mitigating effect. Rapeseed cannot account for more than one third of the total area under annual crops, and is associated with other more favorable crops, such as cereals. This threshold probably mitigates all the potential unfavorable effects observed at the field scale. Maize crops are not subject to such constraints and can dominate the landscape, leading to an absence of such mitigating effects. Furthermore, fixed landscape elements have a greater weighting than crops.

Mitigating effects, such as those described above, are consistent with the hypothesis of simple additive effects of crop areas in the landscape during the breeding season. They may account for absence of expected effects, but not opposite effects, such as that of rapeseed in one context. We can explain this last case only by abandoning the hypothesis of simple additive effects, and considering more complex processes. This reasoning is more speculative and we suggest here three hypothetical processes compatible with our results:

(1) "Remote" effects of the crop extending beyond the crop: Rapeseed crops may interact with neighboring habitats because this crop provides more insects than other crops, as it is more attractive to herbivorous insects and pollinators (*Hebinger, 2013*). However, the scenario in which rapeseed acts as a source of food spilling over into neighboring fields remains theoretical. Studies of the food resources for birds associated with crops (*Stoate, Moreby & Szczur, 1998*; *Cléré & Bretagnolle, 2001*; *Moreby & Southway, 2002*) are scarce and seldom comparable, due to methodological differences.

(2) Delayed effects from winter to the breeding season. Rapeseed is a favorable crop for skylark in winter. *Powolny (2012)* observed it was the most selected crop with alfafa, as its leaves provided a useful source of food during this critical period. For resident populations, this process may have visible effects during the breeding season. A beneficial association between set-aside in winter and bird density in the same area in spring was observed by *Whittingham et al. (2005)* for a resident passerine, the yellowhammer (*Emberiza citrinella*). In line with this hypothesis, the positive response

to rapeseed may be accounted for by cumulative effects throughout the year, whereas field-scale studies generally focus on partial effects during the breeding season. This mechanism depends on the migratory behavior of the skylarks. A resident population would benefit from rapeseed all over the year, whereas a migratory would not. Resident and migrant populations are poorly delimited in continental Western Europe and a mixture of resident and migratory behavior was observed in one population from the Netherlands (*Hegemann et al., 2010*).

(3) Effect of the crop as a function of its area, with positive effects in small areas becoming negative with increasing area size. Quadratic responses of this type may be accounted for by ecological processes, such as 'landscape complementation' (*Dunning, Danielson & Pulliam, 1992*), in small areas, followed by a detrimental loss of appropriate habitats when the crop area exceeds a given threshold. This scenario may be rendered more complex by adding a temporal dimension, as complementation between crops may occur during the breeding season. The growth of rapeseed makes the vegetation structure unsuitable for nesting (*Wilson et al., 1997*), but some studies suggest rapeseed is more suitable in the early season than later. *Eraud & Boutin (2002)* found that skylark density in rapeseed decreased throughout the breeding season, and a positive association between rapeseed and skylark was observed in some cases in early breeding season (*Chamberlain & Gregory, 1999*) or with underdeveloped rapeseed (*Whittingham, Wilson & Donald, 2003*). This could cause skylark to shift to other more favorable crops, as observed in the case of winter wheat (*Chamberlain et al., 1999*; *Donald et al., 2002*; *Hiron, Berg & Pärt, 2012*). According to this hypothesis, the area of rapeseed is less important than crop diversity allowing the succession of suitable crop mosaics on the landscape during the breeding season. Habitat diversity on 1 km$^2$ lanscapes, however, was found to have a negative effect on skylark abundance in UK lowlands (*Chamberlain et al., 1999*; *Pickett & Siriwardena, 2011*). Facing this, *Chamberlain et al. (1999)* questioned the equal weight given to each crop in their diversity index. The solution probably lies in the development of crop diversity indices taking into account the growth dynamics of the crops and not their simple nature.

In conclusion, the effects of crops were not simply additive when switching from field to landscape, but the underlying causal mechanisms remain unclear. If we are to understand such processes, we must take into account subtle interactions between crops, and between crops and fixed elements, and further investigations of the shape of the responses are required.

## Importance of context

We observed contrasts between ecological regions (for rapeseed) and between openfield and grassland contexts (for maize). These sources of variability were expected, but their true origins remain unclear. Possible underlying mechanisms were discussed by *Whittingham et al. (2007)* and *Schaub et al. (2011)*. The reflections of these authors call into question the tendency to oversimplify objects for conceptual reasons (lack of prior knowledge of their variability) or practical reasons (data availability). Indeed, regional

differences may indicate that the bird populations evolved differently, with different habitat preferences (unlikely according to *Whittingham et al. (2007)*), or types of migratory behavior, with consequences mentioned here above. Regional differences may also result from ecological or agronomic gradients that are unknown or cannot be described at the required resolution, e.g., agricultural practices (pesticide use, previous crop, soil tillage) resulting in differences in a given crop between regions, from the bird's point of view. For example, *Shrubb (1988)* showed that, in winter, lapwings (*Vanellus vanellus*) could differentiate between wheat following rapeseed and wheat following wheat, due to the stimulatory effect on the soil fauna of the organic manure applied after rapeseed in the cropping systems of West Sussex. This example highlights the complexity of the agronomic processes potentially affecting crops and subsequent species-habitat associations. The spatial variation of the responses raises a practical problem. In a perspective of applied research, the question is not so much determining whether or not there is an effect, as identifying the conditions and locations in which this effect is expressed. However, it was not our goal. We aimed instead simply to highlight differences, revealing gradients operating at large scales and the influence of some key elements of the landscape.

## Consequences for management

Our results concerning field/landscape inconsistencies and variations with local and regional context may reasonably be assumed to apply to situations other than that of the effect of spatiotemporal crop allocation on skylark. We consider here implications for future studies on both sides of decision-making and local management. Our findings call into question the analytical approaches aggregating the effects of individual habitats in methods for assessing and planning land use over large scales (e.g., life cycle assessment (*Geyer et al., 2010*), land use scenarios (*Brandt & Glemnitz, 2014*)). We need to refine the models to catch possible interactions and non-linear responses. For this purpose, field- and landscape-scale studies are complementary and can be put together in both top-down and bottom-up directions, by constructing a hypothesis at one scale and verifying it at the other. We also need to accept that the explanation "the effects are context-dependent" is unsatisfactory in a perspective of applied research for rural extension. The adaptation of management measures advocated by some authors (*Whittingham et al., 2007*; *Schaub et al., 2011*) implies an ability to define the boundaries of contexts precisely. It is easy to recycle existing administrative entities, but this may be difficult to justify if we are focusing on the bird's viewpoint. We still need to open the fuzzy box of "context," with empirical (mapping the responses) or mechanistic (identifying the underlying causes) methods.

These are programmatic rather than practical considerations. We should stress that our study provides no evidence directly useful for advice and rural extension. Skylark abundance was used as a biological indicator, not as an indicator for management regardless of geographic context. Moreover, if we consider crop allocation as a means of improving the status of farmland birds, the effect sizes obtained were so small that the gain would be minimal for a large range of possible losses (crop allocation suboptimal for gross margin, work organization, agronomy, etc.). By contrast, responses were general and so imprecise that

local improvements based on local diagnosis could not be excluded. Is it better to prescribe the same remedy for all patients, on the basis of imprecise models, or to take time the time to examine each case separately? This debate is beyond the scope of agronomy and ecology.

**Glossary–**The following definitions are not canonical and are limited to the context of this study.

| | |
|---|---|
| **Annual crop** | A crop that completes its cycle in less than one year. Annual crops are also arable crops, but not all arable crops are annual (e.g., alfalfa is grown over a period of more than one year). |
| **Crop allocation** | Decision made annually by the farmer, about which crops to grow in which fields. |
| **Crop rotation** | The succession of annual crops in the same field. It usually, but not always, follows a regular and cyclic temporal pattern. |
| **Cropping block** | One or several amalgamated fields, i.e., not separated by linear features such as roads or ditches. |
| **Cropping pattern** | Combination of the crops in the landscape, described by crop areas (crop composition) and field shape and organization (field configuration). |
| **Cropping system** | The crop rotation and agricultural practices applied to each crop (e.g., soil tillage, fertilizer use and pesticide applications). The cropping system is considered at the field scale. |
| **Field** | Area cultivated with a single crop, usually maintained, with the same boundaries, from year to year. |
| **Fixed elements or structural landscape features** | All types of stable land use over the time of the study (4 years), i.e., forests, hedges, fields with annual crops, permanent crops, etc. |
| **Landscape** | Continuous space consisting of a number of fields and non-agricultural areas. |
| **Monoculture** | Crop succession with a single annual crop. |
| **Permanent crop, denoted "grass"** | Grass, permanent set-aside, fodder crops such as alfalfa in place for more than one year, but excluding ligneous plants, such as orchard trees and grapevines. |

## ACKNOWLEDGEMENT

We thank the volunteer ornithologists for providing the French Breeding Bird Survey data.

### Funding

The authors declare there was no funding for this work.

### Competing Interests

Christophe Sausse is employee at Terres Inovia, the technical center for oilseed crops, grain legumes and industrial hemp.

### Author Contributions

- Christophe Sausse conceived and designed the experiments, performed the experiments, analyzed the data, contributed reagents/materials/analysis tools, wrote the paper, prepared figures and/or tables.
- Aude Barbottin and Philippe Martin reviewed drafts of the paper.
- Frédéric Jiguet contributed reagents/materials/analysis tools, reviewed drafts of the paper.

### Data Deposition

The data on skylark abundance are the property of the volunteer ornithologists who have not given their permission to publish it alongside this article. The raw data will be provided on request to Frédéric Jiguet (fjiguet@mnhn.fr).

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
