# Peer review of "Do the effects of crops on skylark (Alauda arvensis) differ between the field and landscape scales?"

_PeerJ, doi:10.7717/peerj.1097_

## Round 0.1 · original submission · Minor Revisions

Two reviewers agreed that this is an interesting paper and present important finding in an agroecosystem. However the reviewers pointed out several issues that need to be addressed before it can be published. I believe that the authors are able to tackle these issues readily.

Reviewer 1 ·

Basic reporting

PeerJ (2015:03:444:0:0)

Do the effects of crops on skylarks (Alauda arvensis) differ between the field and landscape scales ?

by Christophe Sausse; Aude Barbottin; Frédéric Jiguet; Philippe Martin

General comments:

The manuscript by Sausse et al. challenges the hypothesis that the effects of crops differ between the field and the landscape scale. More precisely, they studied the impact of maize and rapeseed on the abundance of skylarks Alauda arvensis. They show no consistent results according to rapeseed and poorly consistent for maize. However, the study has merit in its objectives. The manuscript presents some interesting data, but a couple of points need attention. It has the potential to be much better and more relevant. I have several major points which I wish the authors to consider. The main detractors from your case (as I see it) are:

1/ The abstract have to be rewritten. Results are insufficiently detailed.
In the Introduction section, small paragraph related to wintering period and habitats selection would be appreciated.
Finally, authors should more accurately describe their assumptions and predictions in the Introduction section. In fact, authors used the term "expected" in main text without giving any real explanation.

2/ Overall, the Discussion section is slightly to speculative and needs revision. As suggested in the last paragraph of the Introduction, results have to be discuss through results obtained by others studied in UK for example.

3/ Some parts of the manuscript are related to general ecology topics (behavioural ecology in crop selection, ...) and other parts are more generally integrated in the fields of agronomy. This shall makes not easy the reading of the MS.

Specific comments:

Line 53 - 55 : I don't really understand the link between the possibility of various trophic resources for birds and the protective capacity against predators. Please indicate quickly positive and/or negative potential effects of crops structure on nesting and protection (i.e. vegetation heights ?).
Line 55 : Please see Butler et al. 2005 Anim. Behav. or Powolny et al. 2014 PlosOne to be more explicit on anti-predation strategies and habitat choice.
Line 56 - 57 : Please clarify demographic gains thanks to resources, food and protection provided by the crop.
Line 68 : Please add references.
Line 69 : Only during breeding season, but commonly aggregative during migration, stop-over and wintering periods.
Line 71 : Agriculture management (or practices) on biodiversity.
Line 81 : What about crop selection during winter. Please add few words on wintering selectivity in skylarks regarding rapeseed or grassland and alfalfa.
Line 87 : What about mechanisms ? Vegetation growth, feeding resource deficit ?
Line 89 - 92 : What can explain these differences observed between studies ?
Line 96 : Again, what about mechanisms ?
Line 113 : Why 4 km². Do you have any information about skylarks territory or home range size ?
Line 122 : Please clarify your predictions. What are your expected results ?
Line 245 - 246 : Expected ?
Line 272 - 273 : How do you explain this difference ?
Line 276- 277 : Why ? I guess that vegetation structure, food resource availability may differ between rapeseed and cereal crops ?
Line 281 : Please change real world by field
Line 345 : What do you mean by landscapes types ?
Line 351 : What are your hypothesis to explain that different population of birds should have different habitat preferences. I don't understand the link with migratory behaviour.

Experimental design

No Comments

Validity of the findings

No Comments

·

Basic reporting

No comments

Experimental design

No comments

Validity of the findings

The data are not provided in the submission reviewed

Additional comments

This is a very interesting paper providing an elegant analysis of existing data to shed light on an important question in agricultural landscape ecology.
It would be interesting to see the marginal effects of the analysed factors on mean numbers of skylarks – the analysis suggests that these would be small after other factors taken into account.
Line 207 should read: “The group below the curve was called “open-field”, and the group above was…..”
Line 234: Should ‘confusion’ be replaced by ‘correlation’?
Line 249: The wording here is a bit unclear.“The influence of sampling on the results was assessed by repeating the analysis 100 times on random samples smaller on the third” - does this mean that each analysis was carried out on two thirds of the data?
Line 281: replace ‘found’ with ‘find’
Lines 283-284: the first sentence in the paragraph does not make sense – it appears that a word or words are missing.
Line 324 insert ‘by’ between ‘for’ and ‘mitigation’.
Line 342 I am not sure that ‘apettent’is an English word, it is certainly not widely used. Could you just way ‘attractive to herbivorous insects and pollinators’?
Lines 360, 363: I suggest ‘complementarity’ rather than ‘complementation’.
Line 386: there is no ‘c’ in Shrubb
Table 3: ‘Square number’ implies ‘the number of the square’ (singular). I suggest ‘number of squares’ for clarity.
Table 3:Average skylark abundance would be more informative than the maximum, as the latter depends on only one value.
Some additional comments are made in the attached Word document

---

## Round 0.2 · accepted · Accept

The manuscript has been appropriately revised as suggested by the 2 reviewers.